# Chemical Compositions of Brown and Green Seaweed, and Effects on Nutrient Digestibility in Broiler Chickens

**DOI:** 10.3390/ani11072147

**Published:** 2021-07-20

**Authors:** Mohammad Naeem Azizi, Teck Chwen Loh, Hooi Ling Foo, Henny Akit, Wan Ibrahim Izuddin, Nurhazirah Shazali, Eric Lim Teik Chung, Anjas Asmara Samsudin

**Affiliations:** 1Department of Animal Science, Faculty of Agriculture, Universiti Putra Malaysia, UPM Serdang, Seri Kembangan 43400, Malaysia; naimazizi83@gmail.com (M.N.A.); henny@upm.edu.my (H.A.); wanahmadizuddin@gmail.com (W.I.I.); aira_hazirah87@yahoo.com (N.S.); ericlim@upm.edu.my (E.L.T.C.); anjas@upm.edu.my (A.A.S.); 2Department of Pre-Clinic, Faculty of Veterinary Science, Afghanistan National Agricultural Sciences and Technology University ANASTU, Kandahar 3801, Afghanistan; 3Institutes of Tropical Agriculture and Food Security, Universiti Putra Malaysia, UPM Serdang, Seri Kembangan 43400, Malaysia; 4Department of Bioprocess Technology, Faculty of Biotechnology and Biomolecular Science, Universiti Putra Malaysia, UPM Serdang, Seri Kembangan 43400, Malaysia; hlfoo@upm.edu.my; 5Institute of Bioscience, Universiti Putra Malaysia, UPM Serdang, Seri Kembangan 43400, Malaysia

**Keywords:** apparent ileal digestibility, apparent metabolisable energy, broiler, brown seaweed, green seaweed, nutrient contents

## Abstract

**Simple Summary:**

This study aimed to analyse the nutritional properties and apparent ileal digestibility of brown and green seaweed on broiler chickens. Proximate content, mineral and amino acid contents were analysed. In addition, the gross energy value of brown and green seaweed was measured. A digestibility trial was conducted to determine the apparent ileal digestibility of seaweed in broiler chickens. Apparent metabolisable energy was determined as well in this study. Birds were fed with 90.30% seaweed-based diet with an indigestible marker. At the end of the feeding trial, birds were euthanised and ileal digesta was collected. Nutrient contents of experimental feed and digesta were analysed, and gross energy was measured. The results revealed that there was no significant difference in the apparent ileal digestibility of dry matter, organic matter, crude lipid and ash contents among the brown and green seaweed-based diets. The findings also demonstrated that the apparent ileal digestibility of crude protein and crude fibre was significantly higher in brown seaweed compared to green seaweed. Nevertheless, no significant difference was observed in the apparent digestibility of metabolisable energy between the types of seaweed.

**Abstract:**

This study aimed to analyse the nutritional properties, apparent ileal digestibility (AID) and apparent metabolisable energy (AME) of broiler chickens fed with brown seaweed (BS) and green seaweed (GS). Proximate analysis was performed to determine the nutrient composition of seaweed. The amino acids were determined using high-performance liquid chromatography (HPLC), and atomic absorption spectroscopy was used to determine the minerals content. The gross energy (GE) was determined using a fully automatic bomb calorimeter, and the AME value was calculated. Titanium dioxide (TiO_2_) was used as an indigestible marker to calculate the AID. A digestibility trial was conducted to investigate the effects of seaweeds on crude protein (CP), crude fibre (CF), ether extract (EE), dry matter (DM), organic matter (OM), amino acids (AA) and minerals digestibility, and AME on broiler chickens. Thirty-six broiler chickens were randomly distributed into two dietary treatment groups with six replicates and three birds per replicate. Results showed that brown and green seaweed was a source of macro and micronutrients. For the AME and AID of seaweed-based diets, the results showed that the AME value for BS and GS was 2894.13 and 2780.70 kcal/kg, respectively. The AID of BS and GS was 88.82% and 86.8% for EE, 82.03% and 80.6% for OM, 60.69% and 57.80% for CP, 48.56 and 44.02% for CF, and 17.97 and 19.40% for ash contents, respectively. Meanwhile, the AID of CP and CF was significantly higher for BS compared to the GS. Findings showed that the AID of various AA was 40.96 to 77.54%, and the AID of selected minerals (Ca, Na, K, Mg, Zn, Cu, Fe) for both BS and GS groups were above 90%.

## 1. Introduction

Seaweed is marine algae that grows in various types of water [1]. Seaweed is originally non-flowering photosynthetic macroalgae that occurs in streaming sections of oceans, seas and rivers [2]. Seaweed is divided into three different groups: brown, green and red seaweed that have been scientifically distinguished based on their colours [3]. Seaweed is a source of macro and micronutrients and different biological bioactive components [4,5].

Digestion contributes to the physical and biochemical breakdown of feed and nutrients in the body in preparation for absorption. Therefore, digestibility is an essential factor showing the utilisation rate of nutritive factors. Generally, poultry feed is formulated based on nutrient digestibility and absorption [6]. In recent years, different marine organisms have been considered valuable biological compounds for livestock [2,7]. Seaweed has numerous bioactive components such as carotenoids, phenolic compounds, tocopherols, peptides and various sulphated and carboxylated polysaccharides such as alginate, ulvan and fucoidan [4]. Sulphated polysaccharides act as antioxidants due to their hydrogen, which combines with radicals and makes it a stable radical to cut off the radical chain reaction [8,9]. Furthermore, the sterols from marine sources have shown anti-inflammatory and cholesterol-lowering activities [3]. The bioactive compounds of seaweed can act as prebiotics and enhance livestock immune response due to various mechanisms [4,10,11]. Additionally, some exogenous substances of the diet able to fix themselves to animal tissues; for instance, polyphenols and similar molecules have been shown to bind to bone [12]. However, most seaweed species have low digestible protein in terms of being an appropriate substitute protein source in livestock feed [13].

Seaweed has different nutrient compositions and bioactive molecules. The composition of seaweed can be influenced by the drying process, harvesting, geography, environmental parameters and seaweed varieties [14,15,16]. Proximate analysis for different species of green, brown and red seaweed has shown that seaweed contained 75.95 to 96.03% moisture content, 26.86 to 74.10% carbohydrate, 4.03 to 34.71% CF, 6.05 to 45.04% ash, 5.22 to 17.28% CP and 0.15 to 0.84% crude fat contents of dry weight [17].

Previous research indicated that seaweed has other health and growth-promoting biological effects [13,18,19,20]. However, there is a lack of research to determine its effects on broiler nutrient digestibility.

## 2. Materials and Methods

### 2.1. Ethics

The study was conducted in the Poultry Unit, Department of Animal Science, Universiti Putra Malaysia (UPM). All animal handling practices were carried out following the guidelines approved by the Institutional Animal Care and Use Committee of the UPM (UPM/IACUC/AUP-R093/2019).

### 2.2. Birds and Experimental Designs

A total of 36 one-day-old male broiler chicks (Cobb 500) were obtained from a local hatchery and raised following the recommended flock management for Cobb 500. The chicks were raised in an environmentally controlled close house, equipped with a penning cage system and plastic mesh flooring. The size of the cage was 120 × 120 cm (length × width). The house temperature was set at 32 ± 1 °C on day 1, then reduced gradually to about 24 ± 1 °C by day 39. The average relative humidity ranged between 60 and 75%. All chickens were vaccinated against Newcastle disease and infectious bronchitis disease (ND-IB) by eye drop at 7 and 21 days of age. The infectious bursal disease (IBD) vaccine was applied on day 14 by eye drop. Birds were fed regular starter and finisher feeds on day 1 to 21 and day 22 to 35, respectively, to meet their nutrient requirement (Table 1). Feed (mash form) and drinking water were provided ad libitum. On day 35, the birds were deprived of feed overnight for about 12 h with free access to drinking water to empty their gastro-intestinal tract from any previous feed [21,22]. The birds had uniform body weights (2045.92 ± 50.44 g) and were randomly allocated into two experimental dietary treatment groups (Table 2) with six replicates in each treatment and three birds per replicate (18 birds/treatment). TiO_2_ (0.3%) was added to the diets as an indigestible marker to calculate the AID [23]. The birds were allowed free access to the experimental feed and drinking water for four days. No mortality was found during the experimental period.

### 2.3. Sample Collection

At the end of the experiment, the birds were euthanised by cervical dislocation in a slaughterhouse. The ileal digesta was collected from the Meckel’s diverticulum to 1 cm before the ileo–caecal junction. The digesta was gently squeezed, flushed with normal saline and pooled in a plastic pillbox. Digesta was immediately stored at −20 °C, dried using a freeze drier and stored at −20 °C for further analysis.

### 2.4. Chemical Analysis

The chemical analysis was performed in seaweed, feed and digesta. The proximate analysis was performed as described in AOAC (AOAC, 1995). Total carbohydrate content was calculated by subtracting the weights of CP, EE and ash contents from 100% [100% − (% CP + % EE + % ash)] [24]. Minerals contents were analysed by ashing the sample at 550 °C for 8 h and then dissolved in 1 N hydrochloric acid. The solution was analysed using Atomic Absorption Spectroscopy (PerkinElmer, Waltham, MA, USA) to determine the selected elements (Ca, Na, K, Mg, Zn, Cu, Fe and Mn).

The amino acids were analysed using High-Performance Liquid Chromatography (HPLC) (Agilent 1100, Agilent Technologies, Inc., Santa Clara, CA, USA). A 900 µL of HCl, performic acid and lithium hydroxide hydrolysed samples were mixed with 100 µL internal amino acid standards and separated using HPLC. All separations were carried out on a ZORBAX Eclipse-AAA (4.6 × 150 mm, 3.5 µm) column (ZORBAX, Santa Clara, CA, USA). The column heater temperature was 40 °C. The column was purged with eluent A (40 mM Na2HPO_4_, pH 7.8) and eluent B (ACN: MeOH: water). The pump was set with a flow rate of 2 mL per minute and 26 min as stop time. The scanning fluorescence detector was set at 340 nm as excitation wavelength and 450 nm as the emission wavelength for all amino acids.

The GE value was determined using a fully automatic Bomb Calorimeter (IKA^®^ Adiabatic C2000, Staufen, Germany). A 0.5 g pellet sample was placed into a quartz crucible, fitted into the decomposition vessel and placed in the bomb calorimeter chamber to combust. The purified oxygen gas was automatically filled, and the bomb was operated as an Isoperibol, and the operation took about 23 min to complete.

The TiO_2_ was determined as described by short et al. [25], the samples were ashed at 580 °C for 13 h, and the minerals were digested using 7.4 M sulphuric acid. The standard solution of TiO_2_ was prepared by boiling 150 mg of pure TiO_2_ in 200 mL concentrated sulfuric acid and was toped up to 500 mL with distilled water to get the final concentration (0.3 mg/mL TiO_2_). Ten concentrations of standard solutions (1, 2, 3, 4, 5, 6, 7, 8, 9 and 10 mL of the final concentration) were prepared, and the sample without TiO_2_ was used as a blank. Sample and standard working solutions were measured by using a spectrophotometer set at 410 nm wavelength.

### 2.5. Calculations

The AID percentage was calculated using the following equation [26]; AID% = 100 − [100 × (% TiO_2_ in feed/% TiO_2_ in digesta) × (% nutrient in digesta/% nutrient in feed)].

The AME was calculated as described by Scott and Boldaji [27], using the following equation: AME (kcal/kg) = GE _diet_ − [GE _excreta_, _digesta_ × (Marker _diet_/Marker _excreta_, _digesta_)]; where: Marker = concentration of titanium dioxide (mg/mL).

### 2.6. Statistical Analysis

The data analysis was conducted using the General Linear Model (GLM) of the statistical analysis system (SAS 9.4, SAS Institute, Cary, NC, USA) by one-way ANOVA. Duncan’s Multiple Range Test was used to compare the significant difference between the treatments at *p* < 0.05. The statistical unit for the variables measured was pooled sample per pen. The statistical model was Y*ijk* = µ + T*ij* + E*ijk*, where; Y*ijk* = dependent variable, µ = general mean, T*ij* = effect of dietary treatment (seaweed), E*ijk* = experimental error.

## 3. Results

### 3.1. Chemical Composition of Brown and Green Seaweed

The proximate analysis and mineral compositions, and GE of seaweed are shown in Table 3. Proximate analysis of BS and GS showed that the CP and EE contents of BS were (*p* < 0.05) higher compared to the GS. The carbohydrate content was (*p* < 0.05) higher in GS compared to the BS. No (*p* > 0.05) difference was observed in DM, OM, CF and ash contents. Meanwhile, no (*p* > 0.05) difference was observed for the GE value among seaweed types. No differences (*p* > 0.05) were observed for the minerals content among the seaweed except for Cu, which was (*p* < 0.05) higher in BS compared to the GS.

Regarding the amino acid contents of seaweed (Table 4), the essential amino acids leucine, threonine, and non-essential amino acids aspartic acid and glutamic acid values were (*p* < 0.05) higher in BS compared to the GS. Meanwhile, the glycine content of GS was (*p* < 0.05) higher than the BS. Isoleucine and valine in GS, proline in BS, histidine, methionine, tryptophan and tyrosine in BS and GS were not detected.

### 3.2. Apparent Ileal Digestibility of Nutrients in Broiler Chickens Fed with Brown and Green Seaweed-Based Feeds

Table 5 shows the AME and AID of DM, OM, CP, CF, EE, ash and mineral contents in broiler chickens fed seaweed. Results showed that the AID of BS and GS based-diets was 88.82 and 86.8% for EE, 82.03 and 80.6% for OM, 60.69 and 57.80% for CP, 48.56 and 44.02% for EE, and 17.97 and 19.40% for ash contents, respectively. Compared to the GS group, the AID of CP and EE was higher (*p* < 0.05) in the BS group chickens. Meanwhile, no difference (*p* > 0.05) was observed in AID of DM, EE, OM and ash contents among dietary treatments. The AME value for BS and GS was 2894.13 and 2780.70 kcal/kg, respectively, whereas no difference (*p* > 0.05) was determined for AME value among seaweed groups.

The digestibility results of Ca, Na, K, Mg, Zn, Cu, Fe and Mn showed that among the selected minerals, Ca had the highest level (96.91, 97.61%) of AID. In contrast, the lowest AID was recorded by Mn (67.64, 73.80%) in BS and GS groups birds, respectively. On the other hand, the AID of Cu was (*p* < 0.05) higher for the GS group (94.64%) compared to the BS (89.94%), while no significant difference was found for AID of Ca, Na, K, Mg, Zn, Fe and Mn among seaweed types.

Apparent ileal digestibility of amino acids in broiler chickens fed with brown and green seaweed-based feeds is presented in Table 6. The result showed that the AID of methionine, proline and serine was (*p* < 0.05) higher in the BS group than the GS group. Nevertheless, the apparent ileal digestibility of arginine was higher (*p* < 0.05) for the GS group than the BS. Meanwhile, no (*p* > 0.05) difference was recorded for the apparent ileal digestibility of leucine, isoleucine, valine, phenylalanine, threonine, glycine, aspartate, glutamate, tyrosine and alanine. Lysine, histidine and tryptophan amino acids were not detected during the HPLC analysis.

## 4. Discussion

### 4.1. Nutrient Contents of Brown and Green Seaweed

Seaweed is rich in essential nutrients, and the nutrients composition of seaweed is highly variable. Findings from the current study showed that the CP content was the major component of dried seaweed. BS and GS contained 59.8 and 55.88% CP, 1.28 and 0.30% EE, 5.78 and 5.19% CF, 29.19 and 34.68% carbohydrate, and 9.7 and 9.14% ash contents, respectively. Other research on various selected seaweed species from different geographic areas reported lower results for CP (5.4−10.11% in BS and 14.6−48% in GS) and higher results for CF (7.2−9.5% in BS and 20.38% in GS) and ash (10.8−42.4% in BS in 20.9 −37.59% in GS) respectively [28,29,30,31,32,33,34,35].

The minerals content of seaweed is higher than that of land plants [36]. This result showed that K was the highest-level element among the minerals detected in brown and green seaweed. The Ca, K, Na and Mg levels (0.13−3.96%) detected were lower compared to the previous reports on various brown and green seaweed species (0.16−5.23%) [31,37,38]. Values of some trace elements such as Mn and Fe were within the ranges previously reported [31,39]. Meanwhile, the trace elements Cu and Zn levels were higher than the previous reports [31,37,38,40]. The Na/K ratios of brown and green seaweed were low (0.061 and 0.064), respectively. Previous research also reported a low Na/K ratio (below 1.5) [29,41]. Therefore, seaweed can help to balance high Na/K ratio diets [29].

However, these results cannot be compared with other studies because the nutrient composition of seaweed is different from species to species. Furthermore, processing methods and environmental parameters can also significantly affect the chemical composition of seaweed [15,16,42,43,44].

### 4.2. Apparent Ileal Digestibility of Nutrients in Broiler Chickens Fed with Brown and Green Seaweed-Based Feeds

The digestibility is determined as the nutrient’s apparent digestibility due to the digestibility value overestimation. However, seaweed digestibility is described by several factors whose efficiency directly depends on the chemical composition (either nutritional or anti-nutritional factors) of the seaweed. In contrast, the chemical composition of seaweed is highly variable due to various factors [43].

The current study showed that the macronutrient crude fat had the highest AID (88.82% and 86.8%) for BS and GS groups. Although the crude fat content of BS and GS (1.28 and 0.30%) is relatively low, lipid digestion depends on the composition of their fatty acid profiles and degree of saturation. Lower lipid digestion is attributed to a higher carbon chain in fatty acids, while double bonds positively affect lipid digestion [45]. It has been reported that most seaweed contains unsaturated fatty acids that have double bonds [46].

The result determined that the AID of CP for BS and GS groups was 60.69% and 57.80%, respectively. Most seaweed species have a low amount of digestible protein for being appropriate as substitute protein source for livestock feed [13]. The cell wall of seaweed is rich in different polysaccharides. It could form a stable complex with their protein and make it inaccessible for proteolytic enzymes; thus, the value of seaweed protein digestibility decreases [47].

The results also demonstrated that the digestibility of CP and CF was poorer in GS group chickens compared to the BS group. The reduction in digestibility may be attributed to the mannan content of GS. Mannan is a major polysaccharide present in different species of GS [48]. Mannan polysaccharide is not hydrolysed in the non-ruminant digestive system. Therefore, the nutrient digestibility could be decreased with increasing mannan in birds’ feeding diet [49,50,51,52]. Besides, different GS species contained insoluble fibres like xylan and insoluble cellulose [13,53]. The insoluble fibres have 1,4 linked xylans with 1,3 linked xylose and a small amount of 1,4- linked glucose. Hence, the presence of such insoluble fibres in GS and their interaction with the proteins could reduce the accessibility of proteins to proteolysis, which may cause a decrease in digestibility [47].

The AID of ash content was 17.97 and 19.40% for BS and GS. The ash digestibility in the seaweed diet could be decreased due to the phytic acid present in seaweed [54]. Phytic acid is considered an anti-nutritional factor that interferes with the digestibility and availability of some trace minerals [54,55]. Consequently, phytic acid might lead to a decrease in the total ash digestibility.

The result showed that the apparent digestibility of BS and GS for CF was 48.56 and 44.02%. On the other hand, seaweed additive (50 g seaweed/head/day) was significantly increased CF digestibility in cows [56]. Digestive enzymes of non-ruminant do not hydrolyse dietary fibres content. Hence, it is considered an anti-nutrition factor for poultry [52,57].

The AID of selected minerals for both BS and GS groups was above 90% except for Mn, which had lower AID. Ca had the highest level (96.91, 97.61%) of AID among the selected minerals, while the lowest AID was recorded by Mn (67.64, 73.80%) in BS and GS groups, respectively. No significant difference was observed for the AID of selected minerals among seaweed types (except for Cu, in which AID was significantly higher for GS than the BS). The findings obtained by Urbano and Goi [58] reported that seaweed increased the consumption of minerals in rats. The availabilities of minerals may have been enhanced due to the proper fermentability of seaweed fibres [58].

The AID of various amino acids was 40.96 to 77.54%. Methionine, proline and serine had significantly higher digestibility in the BS group compared to the GS group. In comparison, arginine recorded significantly higher AID for the GS group than the BS group chickens. However, the apparent digestibility results are varied from different research studies [47]. This could be due to the variation of nutrient compositions of diverse seaweed species and the differences in the methods applying for various digestibility studies. Nonetheless, there is limited published data available on seaweed digestibility in poultry.

## 5. Conclusions

Conclusively, brown and green seaweed is a source of macro and micronutrients and metabolisable energy for broiler chickens. The study suggested that broiler chickens can utilise brown and green seaweed in their diets with higher AID of OM, EE and minerals, moderate CP and amino acids and low AID of CF and ash contents.

## Figures and Tables

**Table 1 animals-11-02147-t001:** Ingredient composition of the starter and finisher diets.

Ingredients (%)	Starter	Finisher
Corn (yellow)	46.0	52.0
Soybean meal (dehulled)	40.0	32.0
Wheat pollard	5.0	6.0
Palm oil	4.0	5.10
L-Lysine ^1^	0.20	0.20
DL-Methionine ^2^	0.40	0.30
DCP ^3^	2.60	2.40
Calcium carbonate	0.80	1.0
Choline chloride	0.20	0.20
Salt	0.30	0.30
Mineral mix ^4^	0.15	0.15
Vitamin mix ^5^	0.15	0.15
Antioxidants	0.10	0.10
Toxin binder	0.10	0.10
Total	100	100
Calculated analysis ^6^		
ME (kcal/kg)	3040.16	3149.82
Protein	21.95	19.06
Fat	5.98	7.19
Fibre	4.34	4.00
Calcium	0.83	0.85
Total phosphorous	1.01	0.94
Available phosphorus	0.50	0.47

^1^ L-Lysine (minimum) 78.8%, ^2^ DL-Methionine 99%, ^3^ dicalcium phosphate, ^4^ mineral mix provided per kilogram of product (mineral mix): selenium 0.20 g; iron 80.0 g; manganese 100.0 g; zinc 80.0 g; copper 15.0 g; potassium 4.0 g; sodium 1.50 g; iodine 1.0 g and cobalt 0.25 g, ^5^ Vitamin premix provided per kilogram of product (vitamin premix): vitamin A 35.0 MIU; vitamin D3 9.0 MIU; vitamin E 90.0 g; vitamin K3 6.0 g; vitamin B1 7.0 g; vitamin b2 22.0 g; vitamin B6 12.0 g; vitamin B12 0.070 g; pantothenic acid 35.0 g; nicotinic acid 120.0 g; folic acid 3.0 g; biotin 300.000 mg; phytase 25,000.0 FTU cobalamin 0.05 mg; thiamine 1.43 mg; riboflavin 3.44 mg; folic acid 0.56 mg; biotin 0.05 mg; pantothenic acid 6.46 mg; niacin 40.17 mg and pyridoxine 2.29 mg. ^6^ The diets were formulated using feed live software.

**Table 2 animals-11-02147-t002:** Ingredient composition of the experimental diets.

Ingredients (%)	Dietary Treatments	
	BS ^1^	GS ^2^
Brown seaweed	90.30	0.00
Green seaweed	0.00	90.30
Palm oil	6.00	6.00
CaCO_3_	1.70	1.70
Salt	0.40	0.40
Vitamin premix ^3^	0.50	0.50
Mineral premix ^4^	0.50	0.50
Choline-Cl	0.30	0.30
TiO_2_	0.30	0.30
Total	100	100

^1^ BS: Brown seaweed. ^2^ GS: Green seaweed. ^3^ Vitamin premix provided per kilogram of product (Vitamin premix): Vitamin A 35.0 MIU; vitamin D3 9.0 MIU; vitamin E 90.0 g; vitamin K3 6.0 g; vitamin B1 7.0 g; vitamin B2 22.0 g; vitamin B6 12.0 g; vitamin B12 0.070 g; pantothenic acid 35.0 g; nicotinic acid 120.0 g; folic acid 3.0 g; biotin 300.000 mg; phytase 25,000.0 FTU cobalamin 0.05 mg; thiamine 1.43 mg; riboflavin 3.44 mg; folic acid 0.56 mg; biotin 0.05 mg; pantothenic acid 6.46 mg; niacin 40.17 mg; pyridoxine 2.29 mg. ^4^ Mineral mix provided per kilogram of product (Mineral mix): Selenium 0.20 g; iron 80.0 g; manganese 100.0 g; zinc 80.0 g; copper 15.0 g; potassium 4.0 g; sodium 1.50 g; iodine 1.0 g; cobalt 0.25 g.

**Table 3 animals-11-02147-t003:** Chemical compositions of seaweed (dry basis).

Nutrient Contents	Seaweed		*p*-Values
	BS ^1^	GS ^2^	
Moisture content%	3.18 ± 0.30	3.80 ± 0.06	0.0897
Organic matter%	87.10 ± 0.29	87.06 ± 0.11	0.9312
Dry matter%	96.82 ± 0.30	96.20 ± 0.06	0.0897
Crude protein%	59.8 ± 0.86 ^a^	55.88 ± 0.23 ^b^	0.0046
Crude fibre%	5.78 ± 0.16	5.19 ± 0.19	0.1041
Crude lipid%	1.28 ± 0.01 ^a^	0.30 ± 0.01 ^b^	<0.0001
Carbohydrate%	29.19 ± 0.88 ^b^	34.68 ± 0.24 ^a^	0.0009
Ash%	9.58 ± 0.14	9.17 ± 0.04	0.0863
Gross energy (kcal/kg)	6171.53 ± 32.15	6150.26 ± 29.81	0.7488
Ca%	0.14 ± 0.01	0.13 ± 0.01	0.7482
Na%	0.18 ± 0.02	0.14 ± 0.01	0.1366
K%	2.96 ± 0.5	2.20 ± 0.03	0.2132
Mg%	0.73 ± 0.13	0.55 ± 0.01	0.2321
Zn (mg. 100 g^−1^)	10.73 ± 0.08 ^a^	8.54 ± 0.56 ^b^	0.0174
Cu (mg. 100 g^−1^)	3.21 ± 1.52	2.63 ± 1.31	0.7865
Fe (mg. 100 g^−1^)	14.67 ± 3.51	11.73 ± 2.75	0.5453
Mn (mg. 100 g^−1^)	13.34 ± 1.24	11.14 ± 0.16	0.1534
Na/K ratio	0.061	0.064	0.6210

^a,b^ Means with different superscripts in the same row indicate a significant difference (*p* < 0.05). Values are expressed as mean ± standard error, *n* = 4. ^1^ BS = brown seaweed. ^2^ GS = green seaweed.

**Table 4 animals-11-02147-t004:** Amino acids content of seaweed (ng ^1^/mg of dry basis sample).

Amino Acids	Seaweed		*p*-Values
	BS ^2^	GS ^3^	
Essential amino acids			
Lysine	13.66 ± 1.33	11.34 ± 1.61	0.3899
Leucine	4.76 ± 0.01 ^a^	2.84 ± 0.36 ^b^	0.0271
Isoleucine	5.87 ± 3.33	ND	-
Valine	2.06 ± 1.03	ND	-
Phenyl alanine	1.53 ± 0.42	2.40 ± 0.68	0.3938
Threonine	24.24 ± 0.56 ^a^	8.41 ± 1.51 ^b^	0.0042
Histidine	ND	ND	-
Methionine	ND	ND	-
Arginine	9.41 ± 0.64	8.32 ± 0.50	0.3093
Glycine	4.18 ± 1.89 ^b^	25.41 ± 1.83 ^a^	0.0150
Tryptophan	ND	ND	-
Non-essential amino acids			
Aspartic acid	4.05 ± 0.38 ^a^	2.57 ± 0.25 ^b^	0.0401
Glutamic acid	12.34 ± 2.06 ^a^	5.70 ± 0.22 ^b^	0.0328
Proline	ND	15.67 ± 4.30	-
Serine	10.13 ± 0.47	8.40 ± 0.39	0.1053
Tyrosine	ND	ND	-
Alanine	3.77 ± 0.44	3.30 ± 0.07	0.4071

^a,b^ Means with different superscripts in the same row indicate a significant difference (*p* < 0.05). ND = Not detected. Values are expressed as mean ± standard error, *n* = 3. ^1^ ng; Nanogram. ^2^ BS = Brown seaweed. ^3^ GS = Green seaweed.

**Table 5 animals-11-02147-t005:** AME and AID of DM, OM, CP, CF, EE, ash and minerals contents in broiler chickens fed seaweed.

Nutrient Contents	Dietary Treatments ^1^		*p*-Values
	BS	GS	
Dry matter%	40.78 ± 0.80	39.07 ± 0.30	0.1155
Organic matter%	82.03 ± 0.64	80.60 ± 0.97	0.2860
Crude protein%	60.69 ± 0.85 ^a^	57.80 ± 0.42 ^b^	0.0380
Crude fibre%	48.56 ± 0.79 ^a^	44.02 ± 1.30 ^b^	0.0409
Crude fat%	88.82 ± 2.49	86.80 ± 1.80	0.5460
Ash%	17.97 ± 0.64	19.40 ± 0.97	0.2860
AME ^2^ (kcal/kg)	2894.13 ± 37.35	2780.70 ± 51.41	0.1488
Ca%	96.91 ± 0.57	97.61 ± 0.22	0.2993
Na%	94.32 ± 0.42	96.30 ± 0.61	0.0559
K%	93.66 ± 1.68	95.26 ± 0.38	0.2628
Mg%	83.72 ± 2.61	87.42 ± 1.12	0.2629
Zn%	95.06 ± 1.75	97.27 ± 0.31	0.2819
Cu%	89.94 ± 0.40 ^b^	94.64 ± 1.38 ^a^	0.0307
Fe%	94.09 ± 1.56	95.56 ± 1.16	0.4819
Mn%	67.64 ± 2.95	73.80 ± 5.77	0.3966

^a,b^ Means with different superscripts in the same row indicate a significant difference (*p* < 0.05). Values are expressed as mean ± standard error, *n* = 6. ^1^ Dietary treatments: BS = 90.30% brown seaweed, GS = 90.30% green seaweed. ^2^ AME = Apparent metabolisable energy.

**Table 6 animals-11-02147-t006:** Apparent ileal digestibility of amino acids in broiler chickens fed seaweed-based feeds.

Amino Acids (%)	Dietary Treatments ^1^		*p*-Values
	BS	GS	
Essential amino acids			
Lysine	ND	ND	-
Leucine	73.17 ± 4.18	59.10 ± 3.19	0.0555
Isoleucine	45.76 ± 8.74	56.81 ± 3.57	0.3068
Valine	62.09 ± 3.61	59.67 ± 7.23	0.7792
Phenyl alanine	63.50 ± 4.23	68.04 ± 9.21	0.7252
Threonine	56.68 ± 3.77	56.19 ± 7.12	0.9779
Histidine	ND	ND	-
Methionine	77.54 ± 6.59 ^a^	54.23 ± 2.97 ^b^	0.0322
Arginine	40.96 ± 2.61 ^b^	63.76 ± 2.85 ^a^	0.0042
Glycine	57.53 ± 3.86	58.04 ± 6.83	0.9514
Tryptophan	ND	ND	-
Non-essential amino acids			
Aspartic acid	60.07 ± 3.14	57.71 ± 6.38	0.7562
Glutamic acid	65.54 ± 2.79	62.60 ± 2.48	0.4761
Proline	71.93 ± 3.73 ^a^	49.21 ± 3.63 ^b^	0.0121
Serine	64.82 ± 2.92 ^a^	56.80 ± 0.79 ^b^	0.0450
Tyrosine	53.97 ± 7.01	57.25 ± 2.41	0.6816
Alanine	55.77 ± 4.95	60.81 ± 6.33	0.5645

^a,b^ Means with different superscripts in the same row indicate a significant difference (*p* < 0.05). ND = Not detected. Values are expressed as mean ± standard error, *n* = 6. ^1^ Dietary treatments: BS = 90.30% brown seaweed, GS = 90.30% green seaweed.

## Data Availability

Not Applicable.

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
