# Peer review of "Chemical Compositions of Brown and Green Seaweed, and Effects on Nutrient Digestibility in Broiler Chickens"

_animals, 2021, doi:10.3390/ani11072147_

Round 1

Reviewer 1 Report

Dear Authors,

Many thanks for this piece of work. I really found the topic of interest for novelty and actuality of finding.

I have minor request of change and a couple of questions that could sound 'troublesome' but it is not, nevertheless those should be taken into account (I could be well not the only one who asks...)

Minor comments:

  • please, prefer nutrient composition  to nutrient components (reported in different parts of the text);
  • please, amend references, they are sometimes inconsistent. 
  • please, try to render your writing style a bit more fluent, as sometimes sentences sound as a list of concepts. Moreover,  several grammar errors were found for inconsistency between subject and verb (pl. vs. sing. and viceversa). 

And my questions are:

  • iodine ???
  • Lysine and Cysteine? ( I saw lysine N.D. in the table...and???).

Thank you if you could clarify.

Author Response

Dear sir,

Please find an attached file for the responses to the inquiries and comments.

Thank you. 

Reviewer 2 Report

The authors intended to evaluate Nutritional Compositions and Digestibility of Brown and Green Seaweed. Overall, the manuscript is presented in an orderly manner and has interesting results. Major concerns arose from the lack of details in materials and methods and nutrient units in diet formulation. 

Please find specific comments to your manuscript in the attached PDF file. 

Author Response

Dear sir,

Please find an attached file for the responses to the inquiries and comments.

Thank you very much.

Round 2

Reviewer 1 Report

Dear authors,

Many thanks for the revision of your manuscript, which I continue to find of absolute interest.  However, some amendments did not fulfill the scope, specially if English grammar is considered. Many grammar errors were found and some of those tried to amend  the already correct forms! Many subject/verb are inconsistent as to sing. vs. plu. forms and euthanize is not correct (cervical dislocation in an abattoir is not euthanization, it is slaughtering!!!).

It is a pity that Iodine was not measured in the trial because it might have helped in the understanding of some phenomena observed in other animal species,  also for birds in future trials. You speak of biologically active substances in seaweed but no chemical determination  was performed in this case, except for approximate nutrients. I am perfectly aware that this is a first feeding trial aiming to explore digestibility performance, however,  to substantiate the use of novel ingredients in the diet of food producing animals some biological explanation is needed in  the introduction. As a minor revision, beyond carefully checking grammar throughout the text, I would suggest to refer to Aldritt et al. 2019 Scientific Reports, on the role of exogenous substances of the diet being capable to fix in animal tissues (bone tissue, specially, with the interaction of elements and AA).

Thank you.

Author Response

Dear sir,

We have dealt with all the comments accordingly. Please refer to the attached file for explanation and action taken for the comments.

Thank you very much.

Loh

Reviewer 2 Report

Good work addressing comments

Author Response

Thank you so much.

Kind regards,

Loh